# Changing the Story: The Evaluation of a Leadership Development Programme for Vulnerable and Deaf Youth in South Africa

Kirsty Bastable [1], Paul Cooke [2], Lou Harvey [3], Victoria Olarte [4], Daleen Casteleijn [5] and Shakila Dada [1,*]

[1]  Centre for Augmentative and Alternative Communication, University of Pretoria, Pretoria 0028, South Africa; kgbastable@gmail.com
[2]  Centre for World Cinemas and Digital Cultures, University of Leeds, Leeds LS2 9JT, UK; p.cooke@leeds.ac.uk
[3]  School of Education, University of Leeds, Leeds LS2 9JT, UK; l.t.harvey@leeds.ac.uk
[4]  Hope and Homes for Children, Wiltshire SP3 4LZ, UK; victoria.olarte@hopeandhomes.org
[5]  Department of Occupational Therapy, University of Pretoria, Pretoria 0028, South Africa; daleen.casteleijn@up.ac.za
*   Correspondence: shakila.dada@up.ac.za

**Abstract:** Vulnerable youth and youth with disabilities are at great risk of not having their rights met. In addition, they face challenges with regard to empowerment and participation in their own lives. Youth development programmes frequently focus primarily on the individual skills of the youth. However, reviews have indicated that for youth to be able to drive change, additional opportunities at community and broader society levels are required. This project sought to evaluate the changes facilitated by the Changing the Story—Leadership Development Programme as implemented in the Youth Accountability and Deaf Inclusion in South Africa project, for both vulnerable and Deaf youth. A longitudinal Q-sort methodology was used to measure the youths' changes in perceptions. The results provided evidence of significant change following the programme, including increases in perceptions of empowerment within the community. Furthermore, although vulnerable and Deaf youth began the programme with differing perceptions of self, community and society, these perceptions were more aligned after completion of the programme. The results and challenges experienced using a longitudinal Q-sort methodology are presented and discussed. Recommendations and limitations are also highlighted.

**Keywords:** Changing the Story; Deaf; disabilities; evaluation; leadership; longitudinal; Q-sort methodology; vulnerable youth

## 1. Introduction

Worldwide, vulnerable youth face challenges in relation to having their rights safeguarded and met (Bexell and Jönsson 2017; The United Nations 1989; United Nations Department of Economic and Social Affairs 2019). This is particularly the case for youth in low- and middle-income countries, where 85 percent of the world's youth population live (Cussen et al. 2012; King et al. 2000; McPherson et al. 2016).

The challenges experienced by vulnerable youth have been identified as often arising due to a lack of participation by the youth in their own life decisions (Patton et al. 2016; Sheehan et al. 2017). This may be attributed to factors which are beyond the vulnerable youth's control and include marginalisation due to age or group membership (Auerswald et al. 2017; Patton et al. 2016). Such marginalisation occurs despite research which has highlighted the benefits of youth participation in their own lives (Chowa et al. 2021).

It has been reported that the combination of developing agency and engagement with others and the environment can actually position youth as key facilitators of change (Maganga 2020; Patton et al. 2016). Yet, although the youth are developmentally positioned

to be agents for change within their environments, they require guided opportunities across a range of environments, including at the level of the individual, community and society, to best engage their skills (Chowa et al. 2021; Patton et al. 2016; Tripon 2022). Such skills include, but are not limited to, communication, engagement and leadership. (Chowa et al. 2021, 2023; Patton et al. 2016; Tripon 2022). For communities in particular, however, recent reviews of programmes aimed at providing such skills to vulnerable youth have highlighted an emphasis on research in high income countries, a lack of inclusion of youth with disabilities, and a focus primarily on the youth themselves with limited engagement with their communities or society at large (Bastable et al. 2022; Chowa et al. 2021, 2023). In addition, such reviews highlight the challenges in the evaluation of the studies identified, such as a lack of specificity and reporting detail in the evaluation processes (Bastable et al. 2022; Chowa et al. 2021, 2023).

One of the programmes identified in the review by Bastable et al. (2022) was that called "Changing The Story" (Harvey et al. 2021). Based on the review, the organisations involved in the implementation of the Changing the Story—Leadership Development Programme (CTS-LDP) sought to apply robust evaluation to the CTS-LDP in the South African context. In addition, they sought to specifically include youth with disabilities. These goals resulted in the implementation of the Youth Accountability and Deaf Inclusion in South Africa (YADIS) Project. This paper presents the results of the evaluation of the YADIS iteration of the CTS-LDP. This paper offers some insights into how the evaluation of empowerment programmes can be performed, particularly with the use of methods that include vulnerable youth in the evaluation.

## 2. Methodology

### 2.1. Aims

The aim of this study was to evaluate the Changing the Story—Leadership Development Programme (CTS-LDP), as implemented within the Youth Accountability and Deaf Inclusion in South Africa (YADIS) project, in terms of the vulnerable youth's perceptions of themselves and their agency, in relation to their communities and the country at large. This main aim was accomplished through the following sub-aims:

1. Analyse and compare the changes in the youths' self-perceptions prior to and following the implementation of the CTS-LDP.
2. Describe the youths' perceptions of themselves and their agency in relation to their communities and the country at large, prior to and after the implementation of the CTS-LDP.

### 2.2. Research Design

The active participation of direct stakeholders (individuals directly impacted by the research) in research has been highlighted in recent years as key to ensuring that research is relevant and actually reflects the views of the participants (Krane et al. 2021). With the participation of vulnerable youth at the very core of this study, it was imperative that the youth were able to report their perspectives of their skills and opportunities. In addition, however, when working with individuals who are particularly vulnerable, the safeguarding of participants must also form a key component of the research process. Safeguarding is defined as "the protection of vulnerable people from harm, whether malicious or unintended, by believing and responding to concerns through a systematic approach" (Changing the Story n.d.)

Based on the need for participation in the research, the Q-method research design was used. The Q-method is a participatory research methodology which can provide qualitative data on subjective topics such as perceptions (Durose et al. 2021; Morea 2022; Richards et al. 2013; Wolf 1978). Although the Q-method has typically been applied in cross-sectional or single-group studies, the application of the Q-method in longitudinal studies has been highlighted more recently as a mechanism by which to statistically evaluate changes in perceptions (Morea 2022).

### 2.3. Ethical Approval and Participant Consent

Safeguarding of the youth began with obtaining ethical approval to conduct this study. This was granted by the University of Pretoria (02595761), The University of Leeds (FAHC 20-081) and the Gauteng Department of Education. In addition, permission was provided by the principal of a school for the Deaf to conduct research at their school. Informed consent was obtained from the caregivers of the youth for them to participate in the program evaluation, while all youth provided their assent to participate in the study.

Following ethical approval having been obtained, a specific safeguarding charter was developed by the organisations involved in the YADIS project (Deafkidz International, Thrive, Hope and Homes for Children/One Child One Family).

The charter addressed the specific safeguarding needs of vulnerable as well as Deaf youth and aimed to ensure that:

> *"all the children and young people participating in this project have the right to feel safe at all times and that the adults guiding the project take the collective responsibility to protect this right."* (Safeguarding Charter 2022)

Specific safeguarding needs for the Deaf youth included the provision of "safe" adult contacts who they could contact, who were accepted and trusted by the Deaf youth and able to converse in South African Sign Language. In addition, contact options which did not require verbal communication, such as whatsapp, were provided.

All staff working with the youth were trained in safeguarding in line with the charter and were required to sign their commitment to upholding the safeguarding of youth throughout the programme. In addition, the youth participating in the programme were also trained in safeguarding, the processes and procedures in place to protect them and how they could report any concerns that they might have in this regard. The charter is available in File S1 in the Supplementary Materials.

### 2.4. Context

The study was conducted in Gauteng, South Africa. The vulnerable youth lived in an informal settlement in Ekhuruleni, while the Deaf youth lived in and around a school for Deaf youth in the South of Johannesburg.

### 2.5. Participants

Participants in this study were purposively selected according to the requirements of the CTS-LDP for YADIS which specified the inclusion of vulnerable and Deaf youth with in the project. The organisations involved in the YADIS project work with vulnerable youth and have identified youth with disabilities as being in need of support, however, as per the reviews of Bastable et al. (2022) and Chowa et al. (2023) the validation of programmes for vulnerable youth is an area which has been neglected, hence the organisations felt that it was important to validate the effect of the CTS-LDP on both vulnerable and Deaf youth before its use could be expanded.

As there is a general lack of clarity of the definition of "vulnerable" (Chowa et al. 2021), for this project, "vulnerable youth" were defined as youth whose cultural group were previously disadvantaged due to Apartheid in South Africa, and who lived in an informal settlement served by a safe park in Gauteng. A safe park is defined as "a safe space run by a drop-in centre/early childhood development centre/after-care centre where children can receive a hot meal, support with homework and there is a safe area for play and sports activities." (The National Association of Child Care Workers 2014).

Deaf youth were specifically identified for inclusion alongside the vulnerable youth due to the high levels of risk experienced by Deaf youth, particularly in relation to a lack of early intervention and language development. In addition, most Deaf youth live in hearing families who may not always be equipped to communicate with them. Finally, one in four Deaf girls is reported to be sexually abused in South Africa (Ward et al. 2018). These factors highlight Deaf youth as being particularly vulnerable and in need of the skills to advocate

for themselves. Deaf youth in this programme were youth who met the criterion for schooling in a school for the Deaf. This criterion indicates that the youth must have a primary permanent moderate-to-profound hearing impairment (Gauteng Education Policy Act 2011).

Caregivers of 26 vulnerable youth and 22 Deaf youth were approached to obtain consent for their youth's participation in the programme. All the caregivers of the vulnerable youth consented to the involvement of their youth in the programme, but three of the youth did not provide their assent to be involved in the programme (12%). Five caregivers (22%) of the Deaf youth did not provide consent for their youth to participate in the programme, and two Deaf youth (9%) did not agree to participate in the programme.

A total of 38 youth (12–21 years of age) participated in the programme from a safe park (N = 23 vulnerable youth) and a special needs school (N = 15 Deaf youth) in South Africa.

*2.6. Materials*

Changing The Story (CTS) is a multi-national programme which aims to help vulnerable youth to participate in the building of inclusive civil societies. The programme has been implemented across 12 countries, with the goals achieved using a Theory of Change approach implemented with arts-based projects.

2.6.1. The Leadership Development Programme

The CTS-Leadership Development Programme (CTS-LDP) aims to empower youth through workshops on various topics, which increase their knowledge of the world and enhance their practical skills. The core of the CTS-LDP is the use of film production as a mechanism through which the youth can experience collaboration, express their views, process challenging life circumstances, experience reflective learning and showcase their capabilities. In addition to film production, workshops are also held on gardening, business and financial literacy, general educational support and sports. The workshops include both discussion and hands-on experiences for the youth. The programme aims to gradually develop confidence and self-belief while building skills which support agency and the application of 'downward accountability' directly by the young people themselves (Bawole and Langnel 2016). Importantly, the programme also provides specific opportunities for the youth to engage with the community and broader society. One such opportunity is the showcase, an event where the youth are provided a platform to show the films they have produced to the community and community leaders. During the CTS-LDP the youth meet three times a week, twice on afternoons during the week, and once on the weekend. The youth are guided through a series of workshops by trained youth facilitators.

For the YADIS iteration of the CTS-LDP (hereafter referred to as the YADIS-LDP), the programme was implemented over a three-month period only. Although a longer implementation period had been envisaged initially, unfortunately, due to COVID-19 affecting the implementation schedule and the withdrawal of ongoing funding for the programme, the timeframe had to be adjusted. The facilitators of the programme were youth leaders employed by the safe parks who had facilitated three previous iterations of the program with other groups of vulnerable youth.

Due to logistical constraints, the two midweek sessions were held with each group separately, but the weekend session was held as a combined Deaf and vulnerable group. The general learning workshops were held during the week, i.e., gardening, business and financial literacy, general educational support and sports. During the weekend sessions the workshops were focused on the process of writing and producing a film. The film production sessions are described below.

The following specific adaptations were made to the programme for the Deaf youth to be able to participate fully, both with the facilitators and with their vulnerable peers.

- The development of the safeguarding charter which specifically catered to the inclusion of vulnerable and Deaf youth with all involved organisations.
- The provision of South African Sign Language (SASL) interpreters throughout the programme. The role of the interpreters was both the interpretation of content pro-

vided and the interpretation of peer interactions during each of the workshops. Two certified SASL interpreters were available at every session or event where the Deaf youth would encounter hearing peers or adults.

- Deaf awareness training for the vulnerable youth, which included discussions on diversity and culture as well as a specific introduction to SASL by a certified interpreter and trainer. For the Deaf youth similar training was provided including discussions on diversity and culture as well as strategies which could be used for interacting and communicating with hearing peers. The combined sessions during the programme were held as follows:

Week 1:

Initial midweek, introductory diversity, culture and communication sessions were held with the vulnerable and Deaf youth separately to discuss the process of working with individuals who were different from themselves and provide strategies which the groups could use to communicate effectively with each other. Following the midweek sessions, the two groups met on the weekend session, where they were given an opportunity to get to know each other while engaging in fun activities such as sports and t-shirt tie-dying.

Week 2:

The second weekend session introduced the concept of a campaign (the aim and process) for film production. For this campaign the youth were asked "what are the challenges that you as youth face in your lives?" The aim of this session was to empower the youth to understand and express their experiences and to use critical analysis to identify resolution strategies for social issues in a format which could be shared. For this session, the medium for the sharing of issues was cartoon production. During the midweek sessions that followed the youth worked on their cartoons.

Weeks 3 and 4:

The third and fourth weeks of the programme saw the youth working on their script and film development. The youth were guided in identifying their goals for the campaign. Following this, they developed a script based on reflections from the cartoon production sessions. The youth then had to identify the resources available to them to implement a campaign using community risk and resource mapping. The youth were responsible for all areas of the film, from creating the storyline to identifying locations which could be used, and what resources were available to support filming.

Weeks 5 to 8:

The fifth to the eighth weeks of the programme were dedicated to film production. The youth were trained to use the filming equipment. After that, the youth both acted in and filmed their script under the guidance of a facilitator.

Weeks 9 to 10:

On completion of the filming, the footage was edited to get it ready for viewing. Editing included the use of subtitles for the films to be accessible to the Deaf. Editing was conducted by the facilitator in consultation with the youth. During weeks 9 and 10 the youth participated in further general workshops relating to gardening and sports.

Following the production of the films, the youth from the YADIS programme were involved in a "showcase" event with other safe parks running the CTS-YLP, where their films were aired.

A detailed description of the CTS-YLP programme is available in the facilitators' manual, which is available on request.[1]

In previous iterations of the CTS-LDP, events involving the community and broader society were also included, for example policy events and community engagement events. However, a change in funding which occurred after this iteration of the YADIS-LDP had already begun resulted in the measurement of the effect of the programme being conducted after the structured youth involvement and showcase event but before a final policy event.

2.6.2. The Research Instrument

The research instrument for this study was the Q-method. In the Q-method, participants sort and rank statements which relate to the concourse (shared knowledge of the field of study). The ranking is based on how true each statement is for the participant, or how it fits with their beliefs (Damio 2018). The individuals' perceptions are then factor analysed in order to identify common and differing perceptions (McKeown and Thomas 2014; Morea 2022). The concourse and the Q-sort for this study were developed following the steps recommended by Watts and Stenner (2012), these are (i) the collection of the concepts; (ii) the selection of the Q-sample; (iii) the formulation of the Q-statements and (iv) the validation of the Q-sample. The concepts for the concourse were identified following a scoping review on the topic of youth development/leadership programmes for vulnerable youth, which identified the key components of programmes. These key components included components relating to the individual, the community and broader society. These key components of programmes were validated by youth leaders from the community and organisations who work with vulnerable youth (Bastable et al. 2022) as the constructs for use in the formulation of the Q-statements. From each of these key components the researchers produced ten statements (30 total) to form the Q-sample. An expert panel of youth leaders from the communities provided input on the clarity of each statement (Watts and Stenner 2012).

Due to nationally weak reading levels in South Africa (Howie et al. 2017), the statements were adapted for readability at a maximum of a grade 4 level and supported with visual aids, namely graphic symbols. The visual aids were placed above the related keywords in the statement to assist the youth who may have lower levels of literacy in understanding the concourse (see Figure 1 below). The full list of statements is available in Table S1 and File S2 in the Supplementary Materials.

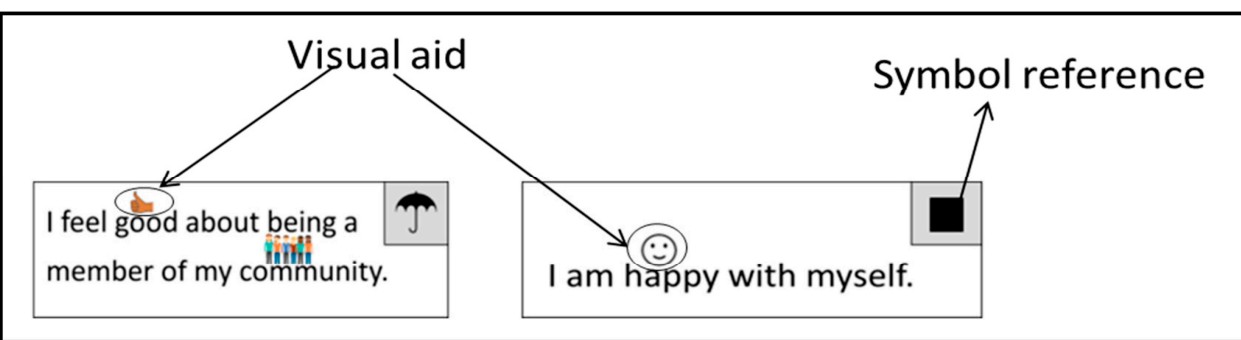

**Figure 1.** Example of two concourse statements as presented to the youth.

In addition to factor analysis, analysis of the concourse included the identification of similar and contradictory statements. There were eight groups of statements in which similar constructs were presented. Prior to the analysis of the actual Q-sorts completed by the youth, the answers provided for the groups of statements were analysed, both on an individual level and by comparing the modes of the statements. Each group of statements was considered in order to identify statements where youth presented contradictory responses. For example, "I can solve my problems" and "Sometimes I get so upset I can't solve problems". If both answers were marked as strongly agreed, a contradiction would arise. The analysis of the grouped statements did not highlight inconsistency in answering, hence all 30 statements were included for the full Q-sort analysis. The contradictory statements analysed are indicated in the list of statements in File S2 in the Supplementary Materials.

*2.7. Data Collection*

2.7.1. Setting and Implementation

The venue used by the vulnerable youth during the week was a demarcated area within the community, regularly used by a safe park. The Deaf youth met in a school classroom. For the combined Saturday group, the venue alternated between a church hall in the vulnerable youths' community and the Deaf youths' school. Transport to and from venues, as well as meals, were provided for participants and their caregivers.

2.7.2. Q-Method Implementation

Prior to the implementation of the YADIS-LDP the participants completed a Q-sort on the concourse statements to produce the pre-programme data. The Q-sort was implemented in the Deaf and vulnerable groups separately, at their own venues.

To begin with, the participants were provided with the concourse statements on a single page of sticker paper with each statement on a separate sticker. The statements were randomly listed. At the top of the page there were also 5 stickers with images of chocolate bars. The youth were then provided with the Q-sort grid. A simplified grid of only 5 options was selected as it was felt to be easier for the youth to make use of (see Figure 2 below) (McKeown and Thomas 2014).

**Figure 2.** The Q-Grid used in this study.

The youth were then asked to look at the chocolate bar stickers and place these in the first row of the grid according to their preference. The process of identifying most liked and least liked; even when it was difficult; was discussed, and linked to the process the youth would need to follow when placing the statements.

The initial sorting of statements was then begun. The Q-method typically includes the sorting of concourse statements on individual cards into three piles: those they agree with, those they feel neutral about and those they disagree with. However, for this study, rather than using three piles of cards, which would be difficult to manage as a group, the participants sorted their statements by highlighting the statements in corresponding

colours. Blue for agreement, yellow for neutrality and orange for disagreement (The highlighter colours also corresponded to the Q-sort grid). Each youth was provided with highlighters for this task.

The statements were read out by the researcher one at a time. A symbol reference was provided in the top right-hand corner of each statement, to ensure that the youth were on the correct statement (a symbol reference was preferred to a number which could be misinterpreted as a ranking). After reading each statement twice the youth were asked to highlight that statement according to how it related to them. Assistance was provided regarding the meaning of statements when requested.

Once all the statements had been sorted into one of the three groupings, placement of the statements into the grid began. The youth were first asked to look at the statements they had highlighted blue (true for me), and pick out the three they felt most strongly about. These were placed in the "this is true for me column". This was repeated for the "not true for me" (orange) statements. The youth were then asked to work through the remaining statements placing them in the grid according to their preference for each statement. The stickers used for the project were able to be lifted and moved as required and the youth were allowed to place statements in a column of a different colour as they sought to complete the grid. Support for reading and understanding of the statements was provided by the researcher and youth leaders throughout this process.

Following the completion of the YADIS-LDP, the Q-sort process was repeated with each group separately to collect the post-programme data. Examples of a completed Q-sort and Q-grid are available in File S3 in the Supplementary Materials.

*2.8. Data Analysis*

The data from the study included pre- and post-YADIS-LDP Q-sorts from both Deaf and vulnerable youth. The data from the two groups were combined and analysed.

The initial analysis of the data was conducted using a paired t-test using SPSS software (IBM Corp n.d.) to compare the youth's perceptions of themselves in relation to the statements prior to and following the implementation of the programme. Each statement as well as each of the key components (individual, community, society) were analysed for significant differences between the youth's perceptions pre- and post-YADIS-LDP.

Further analysis was conducted in line with recommendations for longitudinal Q-method research by Morea (2022). The KADE desktop application (Banasick 2019) was used. Both the Deaf and the vulnerable groups of youth had data which had been incorrectly placed outside of the Q-sort matrix; hence an unforced sort was indicated. Where youth had not placed a statement onto the Q-matrix (10%) a neutral score (0) was applied.

The Q-sorts' factors were analysed using a principal components strategy. Eight principal components were extracted. The factors were then rotated using a varimax strategy. Factor correlation was analysed. Lower factor correlations indicate less commonality between factors. Correlations of greater than 0.8 can indicate problematic correlations (Brown 1993).

Significant factors from the pre-YADIS-LDP were identified. These were factors which had at least two participants with a loading of greater than or equal to 0.7 and an eigenvalue of greater than or equal to 1 (Morea 2022).

The significant factors were described according to how the different groups of participants (vulnerable and Deaf) loaded onto them, as well as by consensus and distinguishing statements for each factor.

For the post-YADIS-LDP analysis, the Q-sorts were again analysed according to a principal components strategy with varimax rotation. The significant factors were then compared to those from the pre-YADIS-LDP, according to eigenvalue, expressed variance, participants loaded, and the consensus and distinguishing statements within each factor. As the factors did not align pre- and post-YADIS-LDP, the movement of participants from the factors identified in the pre-YADIS-LDP Q-sort was described.

A final descriptive analysis focusing on the items which the participants responded most strongly to (i.e., items for which the mode response was either "true for me" [2] or "not true for me" [−2]) was conducted to compare the perceptions of the Deaf and vulnerable youth.

## 3. Results

While evaluating the results from this study it is important for the Q-sort methodology itself to be considered. Specifically, as the Q-sort has a set number of "most agreed/disagreed" areas available within the grid, when a new area of perspective arises for the youth, initially identified areas may be replaced. This process may result in certain areas becoming "more negative" (or positive) within the youths' perceptions, but may actually represent other areas becoming more important, or the current area of focus, rather than the initial areas actually being perceived as less negative. Hence it is important to consider the concourse as a whole, not only as independent statements.

Analysis of the participant's perceptions of the statements of the concourse provided evidence of significant change from pre- to post-YADIS-LDP in seven statements. Two of these statements were at the level of the individual and were rated less true for the participants in the post-Q-sort. Four statements were within the concept of community; two were rated as more true and two less true. The concept of broader society had one statement which showed significant change; this became less true for participants.

No significant difference was identified pre- and post-YADIS-LDP for the key concepts of the individual, community and broader society. The results are available in Table 1 below.

### *Factor Analysis*

The factor analysis of the participant's responses identified one factor to be significantly stronger than the others both pre- and post-YADIS-LDP. Factor 1 provided a 33% and 27% explanation of variance pre- and post-YADIS-LDP, respectively.

Five factors were identified as significant in the pre-YADIS-LDP. These factors showed strong distinctiveness and a significant eigenvalue with low factor correlations recorded. The factor score correlations are available in Table 2 below. The factor array for factor 1 is available in File S4 in the Supplementary Materials. The pre- and post- LDP Q-sort factor analyses are described in Table 3 below.

Of the factors identified in the pre-YADIS-LDP Q-sort, the individual component statement "I am happy with myself" was a consensus statement. The distinguishing statements related to the individual component in factor 3 (Sometimes I get so upset I cannot solve problems), to the community component in factor 1 (I like to be a leader), and to the broader society component in factor 1 (My community listens when I have something to say), 4 (I cannot change government rules), and 8 (It is important for me to know my rights).

In the post-YADIS-LDP Q-sort no consensus statements were evident. Distinguishing statements, however, related to the individual component in factor 1 (I can solve my problems), and the broader society component in factor 3, with two distinguishing statements: some people need more help than others and I cannot change government rules. The remaining factors in the post-YADIS-LDP Q-sort had no individual statements identified as distinguishing.

Of the participants who loaded onto factors in the pre-YADIS-LDP Q-sort, no clear pattern of movement was identified in the post-YADIS-LDP Q-sort. Rather, individual differences appear to have caused the movement. This movement is indicated in Table 4 below.

**Table 1.** Concourse statements, Z-scores and pre/post-YDP difference.

| Key Concept | | Statements | Pre F1 | Post F1 | Mean | CI (95%) | | Significance ($p \leq 0.05$) | |
|---|---|---|---|---|---|---|---|---|---|
| | | (n = 30) | z-Score | z-Score | | Lower | Upper | Statement | Key Concept |
| Individual | 1 | I am happy with myself. | 2.25 | −1.85 | 0.4474 | 0.024 | 0.8708 | 0.039 * | 0.148 |
| | 2 | Sometimes I think I am no good at all. | −0.68 | −0.03 | −0.026 | −0.5 | 0.4478 | 0.911 | |
| | 3 | I do not have anything to be proud of. | −2.1 | −0.94 | −0.184 | −0.622 | 0.2538 | 0.4 | |
| | 4 | I am not good at communicating. | −0.79 | 0.88 | 0.9211 | 0.5139 | 1.3282 | <0.001 * | |
| | 5 | I can solve my problems. | 0.76 | −0.94 | 0.2105 | −0.255 | 0.6764 | 0.366 | |
| | 6 | I am creative | 0.73 | 1.79 | −0.026 | −0.507 | 0.4539 | 0.912 | |
| | 7 | Sometimes I get so upset I can not solve problems. | −0.78 | −0.94 | 0.2105 | −0.262 | 0.6826 | 0.372 | |
| | 8 | I solve problems one step at a time. | 0.91 | −0.94 | 0.079 | −0.413 | 0.5705 | 0.747 | |
| | 9 | I set goals and achieve them. | 1.59 | −0.94 | 0.1053 | −0.371 | 0.5812 | 0.657 | |
| | 10 | I start things but I do not finish them. | −1.32 | −0.94 | 0 | −0.529 | 0.5295 | 1 | |
| Community | 11 | It is hard to work in a group. | −0.6 | −0.03 | 0.1316 | −0.228 | 0.4914 | 0.463 | 0.350 |
| | 12 | I like to be a leader. | 0.98 | −1.85 | 0.4211 | −0.112 | 0.954 | 0.118 | |
| | 13 | When I'm a leader, I listen to people's ideas. | 0.82 | −0.03 | 0.079 | −0.298 | 0.4563 | 0.674 | |
| | 14 | I like to have help from people who are wise. | 0.19 | −0.03 | 0.1842 | −0.304 | 0.6727 | 0.45 | |
| | 15 | People do not understand what I'm going through so they can not help me. | 0.22 | 0.88 | 0.579 | 0.1492 | 1.0087 | 0.01 * | |
| | 16 | I feel good about being a member of my community. | 0.98 | −0.03 | 0.2632 | −0.174 | 0.7001 | 0.23 | |
| | 17 | I can make a difference in my community. | 0.39 | −0.03 | −0.605 | −1.092 | −0.119 | 0.016 * | |
| | 18 | People can learn from my experiences. | 0.28 | 0.88 | −0.605 | −0.968 | −0.242 | 0.002 * | |
| | 19 | I am a peace maker. | −0.31 | −0.94 | 0.0263 | −0.359 | 0.4121 | 0.891 | |
| | 20 | I like working with my community | −0.93 | −0.03 | −0.632 | −1.017 | −0.246 | 0.002 * | |
| Broader society | 21 | My community listens when I have something to say. | −2.26 | −0.03 | −0.158 | −0.696 | 0.3799 | 0.556 | 0.371 |
| | 22 | My community is not treated fairly. | 0.41 | 0.88 | 0.3684 | −0.099 | 0.8359 | 0.119 | |
| | 23 | Social problems affect our lives in my community. | −0.37 | −0.03 | −0.079 | −0.527 | 0.3691 | 0.723 | |
| | 24 | Some people need more help than others. | 0.17 | 0.88 | −0.184 | −0.69 | 0.3219 | 0.465 | |
| | 25 | It is important for everybody to be treated the same. | 0.45 | 1.79 | 0.3947 | −0.006 | 0.7958 | 0.054 | |
| | 26 | I can learn from people who are different from me. | 0.39 | 0.88 | −0.079 | −0.432 | 0.2744 | 0.653 | |
| | 27 | It is important for me to know my rights. | 0.32 | 1.79 | 0.0526 | −0.386 | 0.4913 | 0.809 | |
| | 28 | I can not change government rules. | −0.17 | −0.03 | 0.3158 | −0.168 | 0.7997 | 0.194 | |
| | 29 | My community makes me who I am. | −1.16 | 0.88 | −0.447 | −0.898 | 0.0028 | 0.051 | |
| | 30 | I feel that I can make changes to help my country. | −0.37 | −0.94 | −0.737 | −1.16 | −0.314 | 0.001 * | |

\* $p \leq 0.05$.

**Table 2.** Factor correlations pre-CTS–LDP and their eigenvalue.

| Pre-CTS-YDP factor correlations | | | | | | |
|---|---|---|---|---|---|---|
| | Factor 1 | Factor 3 | Factor 4 | Factor 6 | Factor 8 | Eigenvalue |
| Factor 1 | 1 | 0.3486 | 0.5208 | 0.4236 | 0.5306 | 12.44716145 |
| Factor 3 | 0.3486 | 1 | −0.0323 | 0.2206 | 0.1281 | 2.66246772 |
| Factor 4 | 0.5208 | −0.0323 | 1 | 0.2628 | 0.4098 | 2.25294714 |
| Factor 6 | 0.4236 | 0.2206 | 0.2628 | 1 | 0.1315 | 1.75855919 |
| Factor 8 | 0.5306 | 0.1281 | 0.4098 | 0.1315 | 1 | 1.41034446 |
| Post-CTS-YDP factor correlations | | | | | | |
| | Factor 2 | Factor 3 | Factor 5 | Factor 6 | Factor 8 | Eigenvalue |
| Factor 2 | 1 | 0.3933 | 0.3491 | 0.2914 | 0.1833 | 3.97089226 |
| Factor 3 | 0.3933 | 1 | 0.1788 | 0.2367 | 0.2505 | 3.97089226 |
| Factor 5 | 0.3491 | 0.1788 | 1 | 0.1204 | 0.2355 | 2.53975281 |
| Factor 6 | 0.2914 | 0.2367 | 0.1204 | 1 | 0.3797 | 2.29065586 |
| Factor 8 | 0.1833 | 0.2505 | 0.2355 | 0.3797 | 1 | 1.70309415 |

**Table 3.** Significant factors identified through factor analysis.

| Factor | Explained Variance (%) | Participants Loading [a] (n) | Consensus Statements (Rank) | Distinguishing Statements (Rank) |
|---|---|---|---|---|
| Pre-YADIS-LDP | | | | |
| 1 | 33 | 4V | ○ I am happy with myself. (1) | ○ My community listens when I have something to say.* (30) |
| 3 | 7 | 2D | ○ I am happy with myself. (1) | ○ Sometimes I get so upset I can not solve problems. * (30) |
| 4 | 6 | 3V | ○ I am happy with myself. (3) | ○ I can not change government rules. (29) |
| 6 | 5 | 2D | ○ I am happy with myself. (2) | ○ I like to be a leader. * (1) |
| 8 | 4 | 2V | ○ I am happy with myself. (1) | ○ It is important for me to know my rights. (2) |
| Post-YADIS-LDP | | | | |
| 2 | 10 | 2D; 1V | | ○ I can solve my problems. (1) |
| | | | | ○ Some people need more help than others. (29) |
| 3 | 8 | 1D; 1V | | ○ I can not change government rules. (30) |
| 5 | 7 | 2D; 2V | | |
| 6 | 6 | 2D | | |
| 8 | 4 | 2D | | |

[a] Factor score of ≥0.7; * $p < 0.01$.

Based on the statements in Table 5 which were responded to most strongly, greater consistency can be seen from the Deaf youth, as indicated by 11 of statements which were strongly agreed/disagreed on, in comparison to the vulnerable youth, who had only four statements for which the mode response was strongly agreed (+2) or disagreed (−2) upon. For the vulnerable youth, the same statements were most strongly agreed on in the post-YADIS-LDP Q-sort, while for the Deaf youth only one statement was the same, and the rest were different. For the vulnerable youth, the statements most strongly responded to came from individual (3 statements) and community concepts (1 statement), while the Deaf youth responded most strongly to individual (2 statements), community (4 statements) and broader societal (2 statements) concepts initially, but to community (one statement) and individual (2 statements) concepts in the post-YADIS-LDP Q-sort.

**Table 4.** Participant movement across factors from pre- to post-YADIS-LDP.

| Pre-YADIS-LDP Q-sort Factor | Participant Number | Loading ($p \leq 0.05$) | Post-YADIS-LDP Q-Sort Factor | Loading ($p \leq 0.05$) |
|---|---|---|---|---|
| 1 | 11 (V) | 0.7447 * | 5 | 0.9049 * |
| 1 | 1 (V) | 0.7287 * | 2 | 0.7578 * |
| 1 | 17 (V) | 0.7285 * | 7 | 0.5316 |
| 1 | 23 (V) | 0.7118 * | 7 | 0.6225 |
| 3 | 33 (D) | 0.8609 * | 2 | 0.741 * |
| 3 | 36 (D) | 0.8324 * | 3 | 0.9064 * |
| 4 | 15 (V) | 0.7797 * | 4 | 0.7786 * |
| 4 | 18 (V) | 0.7527 * | 4 | 0.4441 |
| 4 | 13 (V) | 0.7115 * | 5 | 0.5189 |
| 6 | 25 (D) | 0.8061 * | 7 | 0.6444 |
| 6 | 28 (D) | 0.7454 * | 8 | 0.6335 |
| 8 | 8 (V) | 0.8288 * | 1 | 0.652 |
| 8 | 21 (V) | 0.7545 * | 5 | 0.6187 |

* significant ($p < 0.05$).

**Table 5.** Statements most strongly responded to in vulnerable and Deaf groups.

| Pre-Test | n | Post Test | n |
|---|---|---|---|
| Vulnerable Youth (n = 15) | | | |
| I am happy with myself (strongly agreed) | 19 | I am happy with myself (strongly agreed) | 13 |
| I do not have anything to be proud of (strongly disagree) | 16 | | |
| I start things but I don't finish them (strongly disagreed) | 9 | | |
| I like to be a leader (strongly agreed) | 10 | I like to be a leader (strongly agreed) | 9 |
| Deaf youth (n = 15) | | | |
| I am happy with myself (strongly agreed) | 5 | | |
| I start things but I don't finish them (strongly disagreed) | 4 | | |
| I like to be a leader (strongly agreed) | 10 | I like to be a leader (strongly agreed) | 5 |
| I like to have guidance from a person who is wise (strongly agreed) | 3 | | |
| I feel good about being a member of my community (strongly agreed) | 5 | | |
| I like working with my community (strongly agreed) | 2 | | |
| My community is treated unfairly (strongly agreed) | 5 | | |
| I feel that I am able to make changes to my country (strongly agreed) | 2 | | |
| | | Sometimes I think I am no good at all (strongly disagreed) | 5 |
| | | Sometimes I get so upset I can't solve problems (strongly disagreed) | 6 |
| | | I set goals and achieve them (strongly agreed) | 4 |

## 4. Discussion

This study, reporting on the effect of the YADIS-LDP as implemented with vulnerable and Deaf youth, in Gauteng, South Africa, reports significant changes in the perceptions of the participants following the implementation of the YADIS-LDP.

What is evident from the results of the YADIS-LDP is that the youth gained insight into themselves not only as individuals, but as individuals who could guide others (through sharing their experiences) and contribute to their communities. Such changes are in line with other youth development programmes which have identified empowerment and new perspectives as outcomes (Bentz and O'Brien 2019; Schusler et al. 2019). This is a positive outcome for the CTS-LDP as a whole.

For the concourse statements where significant changes were recorded pre- and post-YADIS-LDP implementation, the first statement, "I'm happy with myself", showed a significant decrease in agreement from pre- to post-YADIS-LDP Q-sorts. Although it is possible that some of the youth no longer felt happy with themselves, it is more likely that, having completed a programme which focused on the community, others and facilitating change, the youth were more focused on broader issues than themselves at that point in the process (Bentz and O'Brien 2019; Schusler et al. 2019). Such changes may be considered artifacts of the Q-sort methodology and as such need to be considered within the broader context of the concourse, rather than as absolute changes within individual perceptions. This explanation is supported by the number of items within the concept of community which showed significant change compared to those in the individual or broader societal concepts.

An alternative explanation may be illustrated by the statements that the youth reported most strongly on. "I am happy with myself" was initially most strongly reported by five Deaf youth, however, in the post-YADIS-LDP Q-sort, this statement was not reported as most strongly felt at all. In contrast, in the post-YADIS-LDP, the participants reported on specific skills in which they felt strong. It may be that the YADIS-LDP facilitated skill development for the youth, which enabled them to enunciate the specific elements of happiness that were important to them, rather than the less specific statement of happiness as a whole.

As with much research, the accurate explanation for the changes reported is most likely a combination of both explanations. For example, for the community and broader society concepts specifically, significant positive increases were perceived by the youth with regard to the value of sharing individual experiences with others and working with the community. However, this was also accompanied by increases in perception that others do not understand them and decreases in their perception that they can actually make a change in their community or society at large. These results are somewhat conflicting in nature, but may have been emphasised by the two groups of youth working together where a communication barrier was present. It is possible that where youth shared their experiences and were understood (perhaps by their direct peer group) that this increased their perception of the value of sharing one's experiences, however, if the shared experience was not fully understood or did not result in the desired change, the youth may have felt less empowered. Various studies on the integration of Deaf and hearing youth have emphasised the challenges in establishing social connections which are sufficiently strong for Deaf youth to feel accepted and understood (Lee et al. 2022; Olsson and Gustafsson 2022). In this project, although attempts were made to reduce the possibility of communication breakdowns between the two groups, it is possible that at times neither group felt fully accepted or understood.

Similarly, it is interesting that, pre-YADIS-LDP, the Deaf youth highlighted strong positive feelings of inclusion and working with their community as well as a perception that they were able to make a difference to their country. In contrast, the vulnerable youth did not respond strongly to community/society-related items, ranking them as least important, in spite of the need to know ones' rights being highly ranked.

Post-CTS-LDP, however, the Deaf and vulnerable youth responded more similarly, with both groups having strong positive perceptions about being a leader and being happy with one's self, while Deaf youth also reported strongly on individual skills, such as being able to set and achieve goals and having self-worth.

Feelings of connectedness by the Deaf youth to their community are not surprising, as this is primarily the space in which they are able to communicate in their own language, and the Deaf community plays a specific role in the formation of the identity of Deaf youth (McGuire 2020; Mcilroy and Storbeck 2011; Mildner 2020), which is different from the role of community in the formation of identity for vulnerable youth. Hence, initial positive feelings about the community are unsurprising from the Deaf youth. In the post-YADIS-LDP Q-sort results, similarly to happiness, the both groups of youth appear to be focusing on specific skills which have been facilitated and focused on during the programme, for example, leadership and setting goals. As a result, both groups reported more similar perceptions.

Overall, however, the implementation of the YADIS-LDP resulted in changed perceptions which related to the concept of community experienced by the youth. Facilitating the development of youth involvement in community relates not only to the participation of youth within their communities, but to community sustainability overall, as the youth work with their communities to solve challenges (Franco and Tracey 2019; Suryani and Soedarso 2020).

This study was based on the need for programmes for vulnerable youth to be rigorously evaluated, as specified by Bastable et al. (2022) and Chowa et al. (2023). Furthermore, these reviews highlighted the need for programmes with vulnerable youth to extend beyond the individual to the community and broader society (Bastable et al. 2022; Chowa et al. 2021, 2023). Based on these results the YADIS-LDP is heading in the right direction, however, further emphasis on broader society is still required. In the case of this iteration of the YADIS-LDP, it may be, however, that the funding challenges which were experienced, and which actually resulted in the programme being shortened, impacted the activities which the youth may have otherwise been involved in at a broader society level. For example, the showcase event where the youth presented their films was more of a "farewell" to the programme on this occasion, rather than a staging post in an iterative cycle of project delivery, as it had been in previous iterations of the CTS-LDP. Similarly, data collection had to occur before a policy event that a few of the youth from the programme later attended. Both events could have increased the youth's perceptions of themselves as changemakers in broader society (Schusler et al. 2019).

Finally, the use of the Q-method across groups in a longitudinal study of a Leadership Development Programme is an original research design, which has been shown by this study to be possible, as research moves in a more participatory direction and participants are seen as experts in their fields rather than objects to be studied. Participatory methodologies such as the Q-method have an important role to play in providing mechanisms by which participants without academic knowledge are able to express their knowledge, and how subjective perceptions on a topic can be objectively analysed.

Overall, the implementation of the YADIS-LDP appears to have shifted the perceptions of the youth in relation to specific areas. Such changes in the youths' perceptions speak to a broadening of their perspective, with increased awareness of others and their needs. This broadening has been shown in other programmes to impact career decisions as well as building the capacity of the youth to impact social change beyond the programme (Nicholas et al. 2019; Schusler et al. 2019). Despite these changes, significant results were not evident across the key concepts of individual, community and broader society. It is unfortunate that the data from this study do not point to why this may be the case.

*Limitations*

This study had several limitations, including within the use of the Q-sort. Although this method provided rich information on the perceptions of the youth, the application of this in a group setting created challenges, particularly with regard to not all the youth filling in the Q-sort appropriately. In addition, artifacts of the Q-sort methodology as used in a longitudinal study emerged in that items initially perceived as important/not important showed significant change, possibly not due to change occurring in those items, but more as a result of other items becoming more or less important. Although two sign language interpreters were available at each combined session, this was unfortunately not sufficient for all the youth to be able to interact with each other spontaneously. Finally, the number of participants from the vulnerable youth group was significantly impacted by the post-YADIS-LDP Q-sort being conducted during the school holidays, as they were not always able to attend.

## 5. Conclusions

In conclusion, the YADIS-LDP was able to produce significant changes in specific perceptions of the youth involved. Although the two groups of youth began their journey with differing perceptions, their post-programme results indicate that they now share perceptions to a greater extent. The YADIS-LDP can be seen to have provided changes in their perception, particularly in relation to statements linked to community. However, challenges with the implementation of the programme may have stifled changes in perceptions relating to broader society. The findings relating to the implementation of the Q-sort methodology in a longitudinal study include the ability to implement the Q-methodology in a low-literacy, group environment. However, the analysis of longitudinal results requires the consideration of changes in relation to the concourse as a whole, due to items having to move to accommodate items which have changed in perception, even if the original items are still considered important/not important.

**Supplementary Materials:** The following supporting information can be downloaded at https://www.mdpi.com/article/10.3390/socsci12110631/s1, File S1: Safeguarding Charter; File S2: Concourse Statements. (Westcott 1991; Lomas and Johnson 2012; Sullivan et al. 1987; Kvam 2004; Ridgeway 1993); File S3: Q sort statement page and Completed grid; File S4: Pre-YADIS-LDS Factor 1 Array.

**Author Contributions:** Conceptualization, K.B., P.C., L.H., V.O. and S.D.; Methodology, K.B. and S.D.; Investigation, S.D.; Data curation, K.B. and D.C.; Writing—original draft, K.B. and S.D.; Writing—review & editing, K.B., P.C., L.H., V.O. and D.C.; Supervision, S.D.; Project administration, P.C.; Funding acquisition, P.C., V.O. and S.D. All authors have read and agreed to the published version of the manuscript.

**Funding:** This research was funded by GCRF grant ref. AH/V011626/1; Hope and Homes for Children.

**Institutional Review Board Statement:** The study was conducted in accordance with the Declaration of Helsinki, and approved by the AHC committee on 24 June 2021 and the Ethics Committee of University of Pretoria (HUM037/0821) on 16 March 2022.

**Informed Consent Statement:** Informed consent was obtained from all participants involved in the study.

**Data Availability Statement:** The data presented in this study are available on request from the corresponding author.

**Conflicts of Interest:** The authors declare no conflict of interest.

## Notes

[1]  Professor Paul Cooke, P.Cooke@leeds.ac.uk.

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
