# Peer review of "Changing the Story: The Evaluation of a Leadership Development Programme for Vulnerable and Deaf Youth in South Africa"

_socsci, doi:10.3390/socsci12110631_

Round 1

Reviewer 1 Report

Comments and Suggestions for Authors

Dear authors,

Thank you for the interesting piece. Please find below some comments for your consideration:

Abstract
Recommendations/implications can be added to the abstract

Introduction
In the conclusion portion, it would be beneficial to add information on why the study is essential and what it can contribute to the international social sciences literature.

Methodology
Providing more information about the project as the study context will add value to the paper. This could be presented as a sub-section in this section or as a sub-section in the introduction. Materials sub-section could be re-described as context and materials, and could be presented at the beginning of the methodology section or as a sub-section under the introduction.

Page 3, first sentence - … specifically addressed crucial or any other word to replace specific…

Results
The results could have been more precise if structured in line with specific research questions answered through the presented findings. So, clarifying the questions after the study’s aims or in the introduction section may be helpful.

Discussions
The discussion was well done but could benefit from a revision if the results section is slightly modified.

Conclusions

  • Key conclusions from the study
  • Limitations and Future Studies
  • Implications and lessons learned

I hope you find the feedback beneficial.

All the best!

Author Response

Dear Reviewer,

Please find a point by point to the comments. Thank you for the valuable feedback. 

Thank you for the interesting piece. Please find below some comments for your consideration:

Thank you . 

Abstract
Recommendations/implications can be added to the abstract

A sentence is added. 

Introduction
In the conclusion portion, it would be beneficial to add information on why the study is essential and what it can contribute to the international social sciences literature.

We have added a sentence on the implications for the literature as well. 

Methodology
Providing more information about the project as the study context will add value to the paper. This could be presented as a sub-section in this section or as a sub-section in the introduction. Materials sub-section could be re-described as context and materials, and could be presented at the beginning of the methodology section or as a sub-section under the introduction.

Thank you - There was some contextual information with in the methods. We have now put this under the heading contexts in the methodology to highlight the section more clearly. 

Page 3, first sentence - … specifically addressed crucial or any other word to replace specific…

we removed specifically 

Results
The results could have been more precise if structured in line with specific research questions answered through the presented findings. So, clarifying the questions after the study’s aims or in the introduction section may be helpful.

We have not added a research question but hope that the changes with the results will allow more scpefic descriptions in relations to the aims. 

Discussions
The discussion was well done but could benefit from a revision if the results section is slightly modified.

We trust that this has been adequaltely addressed in the revisions of the results. 

Conclusions

  • Key conclusions from the study
  • Limitations and Future Studies
  • Implications and lessons learned

The discussion contains some of this information. To due word limitations we could not add more to this section. 

I hope you find the feedback beneficial.

Yes, thank you for the valuable feedback. We trust it has improved the quality of the manuscript. Thank you 

All the best!

Reviewer 2 Report

Comments and Suggestions for Authors

I would note that this article is really very interesting, but I would suggest that the design of Figure 1 be cleaned up. It now gives a feeling of disproportion.

I don't have much else to comment on, because everything is clearly described.

Author Response

Dear Reviewer 

I would note that this article is really very interesting, but I would suggest that the design of Figure 1 be cleaned up. It now gives a feeling of disproportion.

Thank you for the feedback We have attempted to improve the quality of the figure. 

I don't have much else to comment on, because everything is clearly described.

Thank you for the positive feedback 

Reviewer 3 Report

Comments and Suggestions for Authors

Good work with some recommendations:

1. The title is very attractive and well written. Good work!

2. In the introduction chapter would be important to insert some other research related to the topic, updated :

Tripon, C. Supporting Future Teachers to Promote Computational Thinking Skills in Teaching STEM—A Case Study. Sustainability 202214, 12663. https://doi.org/10.3390/su141912663

OECD (2023), "Consumer vulnerability in the digital age", OECD Digital Economy Papers, No. 355, OECD Publishing, Paris, https://doi.org/10.1787/4d013cc5-en.

3. How was the research methods used for selecting people involved? Please write some details : it was selected by any criteria, it was randomized research population?

4. How about the research instruments? Describe please some details about how instruments was constructed, pre-tested, the process and the final revision to used in research.

5. Please include some arguments related to the originality and the research newest results related to the sustainable communities.

Author Response

We have attached a response to the comments. Thank you. 

Round 2

Reviewer 3 Report

Comments and Suggestions for Authors

Good manuscript, also recommendations are underline:

1. How was the research methods used for selecting people involved? Please write some details : it was selected by any criteria, it was randomized research population?

2. Please include some arguments related to the originality and the research newest results related to the sustainable communities.

Author Response

Thank you for the opportunity to respond.

  1. How was the research methods used for selecting people involved? Please write some details : it was selected by any criteria, it was randomized research population?

We have added some clarification of this on page 3. We also described how the samples was selected and contextual information. 

2. Please include some arguments related to the originality and the research newest results related to the sustainable communities.

We have added this to the discussion section. Thank you